# Investigating Aerobic Hive Microflora: Role of Surface Microbiome of Apis Mellifera

**DOI:** 10.3390/biology14010088

**Published:** 2025-01-17

**Authors:** Grigory Kashchenko, Amir Taldaev, Leonid Adonin, Daniil Smutin

**Affiliations:** 1Faculty of Geology, Soil Science and Landscape Studies, Russian State Agrarian University—Moscow Timiryazev Agricultural Academy, 127550 Moscow, Russia; 2Dokuchaev Soil Institute, 119017 Moscow, Russia; 3Shemyakin-Ovchinnikov Institute of Bioorganic Chemistry, 117997 Moscow, Russia; 4Research Center for Molecular Mechanisms of Aging and Age-Related Diseases, Moscow Center for Advanced Studies, 123592 Moscow, Russia; 5Federal State Budget-Financed Educational Institution of Higher Education, The Bonch-Bruevich Saint-Petersburg State University of Telecommunications, 193232 St. Petersburg, Russia; 6Faculty of Information Technology and Programming, ITMO University, 197101 St. Petersburg, Russia

**Keywords:** *Apis mellifera*, 16S, whole-metagenomic sequencing, bacterial diversity, cuticular communities, symbiosis

## Abstract

Honeybees host various microbes on their exoskeleton, and this study examined how these microorganisms influence bee health and behavior. Using advanced techniques, we identified distinct bacteria and fungi on the bees’ surfaces, which differ from those found in their guts or hives. Notably, we found increased levels of Actinobacteria on the bees’ surface, but the origin and role of these microbes remain unclear. This research highlights the complexity of honeybee microbiomes and suggests that surface microbes may play a protective role against diseases. Future studies should investigate how these microbial communities change over time, particularly in the fall when bees are more susceptible to illness. A deeper understanding of these dynamics could inform novel strategies for bee conservation.

## 1. Introduction

The cutaneous microbiome exerts a significant influence on host biology. While extensively studied in humans and vertebrates, research on invertebrate surface microbiota remains comparatively limited [1,2]. This disparity is likely due to two key factors: the greater stability of gut microbiomes compared with cutaneous microbiomes, facilitating less-contaminated sampling, and the inherently low biomass associated with the insects’ exoskeletons [1,3], which pose challenges to cuticular microbiota identification.

Perhaps the relatively low level of cuticular microflora is related to its importance in the mating process: arthropod cuticular microflora influences mating behavior [4,5], which could theoretically lead to sympatric speciation [5]. The insect cuticle carries large amounts of hydrocarbons (CHCs), which play a crucial role in defense against pathogenic microflora, cuticle lubrication, adhesion effects linked with walking [6,7], and even mating or colony members recognition [8]. Cuticular microorganisms may influence CHC profiles, which, on the one hand, creates a barrier to the settlement of this environment and, on the other hand, can maintain the constancy of the surface microbiota. Besides CHCs, on the insects’ body surfaces, a wide spectrum of antimicrobials against “micro-invaders” (AMPs, aminoglycosides, oligosaccharides, benzoquinones, etc.) can be found [9,10]. They can both prevent the settlement of this substrate and the penetration of pathogens, maintain the stability of the cuticular community, and even be partially produced by it. Meanwhile, the chitinous cuticle of terrestrial arthropods is likely a poor substrate for microorganisms. Therefore, it is promising to look for specific cuticular microorganisms on the surfaces of social arthropods that constantly interact with rich substrates. In such a case, part of the barrier function of the cuticle may be performed by microorganisms on its surface, using various mechanisms to prevent the invasion of pathogens.

The cuticular microbiota of a variety of terrestrial arthropods [1], including *Drosophila* [3,5,11], spiders [4,12], ants [13,14], and, recently, bees [15,16,17], has been the subject of documented research. The available evidence suggests that these microbiomes can influence host behavior [4,12], thereby highlighting the critical role of surface microbiota regulation in host–microbe coevolution. Studies on ants demonstrate a correlation between body size and both the biomass and diversity of associated bacterial communities [13,14], indicating possibilities for higher contamination, the development of new microenvironments, or some methodological issues. In general, the cuticle acts as a key barrier between the host and the external environment, and the cuticular microbiome influences this barrier function [1,18,19].

While the honeybee gut microbiota has been well studied [20], the role of aerobic microorganisms in other aerobic hive habitats, including the cuticle, remains under debate [2,21,22]. The honeybee’s hive environmental microbiota is believed to significantly impact pollen and bee bread processing, preservation, and, ultimately, the nutritional quality of bee products [2,15,16]. However, limited comprehensive metagenomic studies reveal that less than 1–2% of the bacterial population differs between the whole-body and gut microbiome [23]. This means that if the gut bacterial load is about 10^8^–10^9^ [20], the bacterial load of the whole-body surface might be 10^7^–10^8^. While this suggests potential variations in the microbiota associated with other body regions, previous attempts at sequencing honeybee tissues excluding the gut have yielded inconclusive results.

The bee gut microbiota elucidates the composition of the honeybee’s cuticular microbiota, and researchers have employed both cultivation-based and molecular techniques [15,24,25]. Recent studies on the cuticular microbiomes of Hymenoptera reveal distinct profiles between social and solitary bees. Solitary bees show greater microbial diversity, while honeybees have a more stable microbiota, reflecting their ecological roles. The high social interactions in honeybee colonies, such as allogrooming and co-cleaning, enhance microbiome stability, which partially overlaps with their gut microbiota. Studies show that predominant bacteria groups on the honey bee surface include some “gut” species (*Lactobacillus*, *Bifidobacterium*, *Gilliamella*, *Bombella*, and *Snodgrassella*) and some possibly unique species for this substrate (*Bacillus*, *Fructobacillus*, and *Rhizobiales*) [15,25]. These peculiar microorganisms, commonly associated with plant substrates, play a significant role in honeybee metabolism, particularly carbohydrate fermentation and utilization, and immune function [26,27]. The fungal component of the cuticular microbiota includes *Aureobasidium*, *Debaryomyces*, *Alternaria*, *Capnoidales*, *Metschnikowia*, *Saccharomyces*, and *Melampsora*. While many are plant pathogens, yeasts are also found alongside lactic acid bacteria in fresh honey and bee bread. The use of *Aureobasidium pullulans* in biocontrol against plant pathogens [25,28] suggests a potential role of these microbial species in bee health and productivity.

This study addresses the knowledge gap regarding the honeybee surface microbiome by characterizing its composition and potential functions using both cultivation-based and molecular (16S rRNA and whole-metagenome sequencing) techniques. We aimed to develop methods for effective cuticular microbiome research and elucidate the diversity, distribution, and function of this microbiome across different body parts, building upon previous research demonstrating the influence of the surface microbiota in other arthropods and the limited existing data on honeybees. Our investigation explores the hypothesis that this understudied microbiome plays a significant role in honeybee health and metabolism and potentially contributes to colony resilience through interactions with both environmental and gut microbiotas.

## 2. Methods

In this research, we attempted to isolate microorganisms from the surface of winter bees, evaluated interactions between some microorganisms, and confirmed their presence on the surface using metagenomics. Cultures were isolated from the surfaces of bees, and then pure cultures were obtained. Potential antibiotic producers were tested for the release of antimicrobial compounds on different media. The effect of the production of these compounds on other pure cultures was evaluated. To determine the taxonomic composition of the surface, sequencing of flushes from each bee’s surface was performed after enrichment on a liquid medium. In this article, the names of the cultures correspond to the Appendix A.

### 2.1. Culture Methods

The standard media employed (GRM, GMF, Czapek, Sabouraud, Soy, A4, Gauze II; see Appendix A) comprised both liquid and solid (20% agar) preparations. The bees (collected on 1 December 2021 from three hives) were stored individually at −20 °C. To isolate cuticular microbes, 2, 2, and 1 winter bees from different hives (5 in total) were held for one hour in sterile containers (150 mL), rinsed with 1 mL of water, and plated directly or after pre-enrichment in Czapek’s medium. Flushes were obtained by sequentially processing body parts (with 300–1200 µL of wash/media). To avoid contamination, the wings, limbs, and cuticle of the head capsule, thorax, and abdomen without the last two segments were separated under the binocular. After that, we vortexed the solutions (1 min) and inoculated them onto solid media (2–10 days of incubation). Following this, the enriched samples were plated in order to isolate pure lines. Further experiments were then conducted; their design is described below.

All obtained isolates were cultured on appropriate solid media. Their morphology and cellular structure were determined in unstained preparations and, if necessary, stained with methylene blue or Lugol’s solution and Gram and Ziehl–Neelsen stains. If all parameters coincided in isolates from the same source, they were considered identical, and then only one of the lines was cultured, preserved, and analyzed.

We hypothesized that the contaminating microorganisms would probably both readily enter the surfaces of the bees and readily fall off the surfaces. For the experiments on “shedding” from the cuticle surface, inoculations were performed from initially sterile surfaces with which the bees were in contact. Five bees were kept for one hour in sterile 150 mL containers. The containers were then rinsed with 1 mL of water, which was transferred to a solid medium directly or first enriched in 10 mL of Czapek’s nutrient medium.

A series of experiments were performed to select the optimum conditions for the development in the elective medium and to select techniques for flushing the microbiomes. After adding 1200 µL of water to each sample, the sample was vortexed for one minute, after which the flush and the sample itself were placed into a series of elective media (10 mL each) with a varying pH from 6 to 8 in 0.2 increments. Every 24 h, 1 mL of the elective medium was transferred to solid media for the further separation of pure lines. Three different compositions of washing solutions were used: (1) distilled water; (2) 0.9% NaCl in water; and (3) 0.001% SDS & 0.9% NaCl in water.

We hypothesized that different bacteria might be on different parts of the bees, as these parts interact differently with the substrate, and also the bacteria might affect the recognition of individuals or be shed during flight. To determine the differences between the microbiota of the different parts of the surfaces, the bees were cut with sterile needles, the intestines were removed, and their microbiotas were analyzed separately to exclude contamination. Identical lines from the surface and from the intestines were not considered. The samples were placed in 300 μL of 0.9% NaCl solution, vortexed for one minute, and placed in 10 mL of a series of elective media with and without antimycotics added. The media were placed on a rocker for a week, and from day 3 to day 15, 200 µL of each medium was transferred to solid media every other day to obtain isolates.

Tests for the production of antimicrobial compounds were performed. For the dot test, the test culture was transferred to the center of a Petri dish, and the inhibiting culture was transferred in 4 symmetrical dots at a distance of 2.5 cm from the center. For the line test, the suppressed culture was transferred along the diameter with a continuous stroke using a 2 mm thick glass rod. Perpendicular to it, the inhibitory culture was applied with 4–5 parallel strokes with a 2 mm glass rod. The size of the suppression zone was defined as the minimum distance between the inhibiting and suppressed cultures. For the CFU count and suppression zone measurements, ImageJ was used [29].

### 2.2. DNA Extraction and Preparation

For the metagenomics and 16S quantifications, 10 bees were collected from three hives. After washing, the samples were enriched on Czapek’s nutrient medium for 3 days and then sequenced. DNA isolation from the isolates was performed using the DNeasy Blood & Tissue Kit (QIAGEN, Hilden, Germany) according to the protocol recommended by the manufacturer. The DNA library was quantified and quality-assured with a capillary electrophoresis TapeStation 4200 (Agilent, Santa Clara, CA, USA). For 16S rRNA sequencing, we used the Quick-16S NGS Library Prep Kit (Zymo Research, Irvine, CA, USA). 16S rRNA sequencing was performed using Illumina MiSeq (San Diego, CA, USA), with an expected read length of 250 bp. WGS sequencing was performed using Illumina HiSeq (San Diego, CA, USA), with an expected read length of 100 bp.

### 2.3. Metagenomic Analysis

16S amplicon sequencing data analysis was performed using QIIME 2 [30] (version 2024.5.0). Following the import, the DADA2 module [31] was applied for data denoising. Taxonomy was assigned using naive Bayes classifiers with the SILVA [32], GreenGenes [33], and BEExact [34] databases. A phylogenetic tree was constructed using the IQ-TREE [35] plugin (version 2.3.4). Metabolic annotation was carried out using the PICRUSt2 [36] plugin (version 2.5.3).

WGS data trimming was performed using fastp [37] (version 0.23.4), with a minimum quality threshold of Q20, a preference for front trimming, and a sliding window of size 10. For metagenomic annotation, a comprehensive analysis was conducted using Kraken 2 [38] (version 2.1.3) with Bracken [39] (version 2.9) correction on the pre-built NCBI nucleotide database (nt). To annotate potential fungal metagenomic sequences, Kaiju [40] (version 1.10.1) was employed with the pre-built RefSeq fungal sequences index.

To eliminate possible contamination, we compared our data with sequencing data from the same hive’s combs [41] (BioProject PRJNA1048732) and with pan-metagenomic data [15] (BioProject PRJNA879967). Host and human reads were removed using Bowtie 2 [42] (version 2.5.2) with the ‘very-sensitive-local’ preset. The genomes for dehosting and decontamination are available in NCBI GenBank under the following IDs: GCA_003254395.2, GCA_029169275.1, GCA_000184785.2, GCA_000469605.1, GCA_014066325.1, and GCA_000001405.29.

Metagenomic assembly was performed using metaSPAdes [43] and IDBA-UD [44] (version 1.1.3). After quality control with MetaQUAST [45] (version 5.2.0), IDBA-UD was selected with a k-mer step of 20 and k-mer sizes ranging from 16 to 116, which produced maximal N50. The PROKKA [46] tool (version 1.14.5) was then used to perform metabolism reconstruction on the selected assembly.

For data visualization, we used R [47] with the tidyverse [48], vegan [49], ape [50], and aRchiteuthis libraries [51].

## 3. Results

The cultivation conditions and media were optimized through experimentation utilizing a range of sample sources.

Keeping the bees in a sterile medium and further sowing the flushes onto solid media yielded surprisingly low CFU counts (Figure 1(a1)), so at least the cultured microorganisms were quite tightly bound to the cuticle of the winter bees and did not fall off easily. Sowing the flushes directly onto the nutrient media led to the detection of relatively small CFUs of fungi in the medium (Figure 1(a2)), suppressing the activity of other microorganisms and resulting in no real cell counts in the medium. The highest biodiversity was achieved after pre-cultivation on an enrichment liquid medium (Appendix A).

To account for organisms not densely attached to the surfaces of the bees, the microbiotas obtained from the walls of the containers in which the bees were transported were analyzed. Only *Aspergillus* and *Penicillium* were detected on Soburo’s medium after direct bee washing; other species were cultured on Czapek’s medium. The composition of organisms obtained on Czapek’s, GRM, and GMF media (Appendix A) did not differ.

Seeding from the abdomen, limbs, and cuticular hairs was also performed to optimize the technique (Appendix A). When using a solution with a mixture of compounds to disrupt biofilms, cultures could not be isolated (Appendix A). The highest diversity was obtained in the cultures on solid nutrient agar after enrichment on a liquid nutrient medium for 24 h, isolating 100–5000 CFUs from different bee organs (Appendix A).

Different cultures with complex morphologies, which were often not preserved during passages, as well as mixed colonies consisting of cells of different morphologies and probably from several taxa, were frequently observed in the cultures (Appendix A). In total, more than 250 colonies were obtained from the cuticles of 10 adult worker winter bees, of which 54 differed in their morphological and physiological criteria (Appendix A). Of these, only a small fraction (no more than 20%) were also readily isolated from the hive environment. Cultures S5c, S5d, and S6d (*Aspergillus* spp.) and S6a (*Penicillium* sp.) dominated the cultures from dry dead bees from the same hive.

Two different cultures were obtained that inhibited the development of other microorganisms (Figure 1b). Cultures similar to S10b were independently isolated from 4 out of 10 bees tested from two out of three different hives. Culture S10f was isolated only once from a sample in which S10b was also present.

Only the culture of the unknown Actinomycete S10b retained antimicrobial activity when transferred to other media. The highest zone of suppression was observed on Czapek’s medium, while on A4, soy, or Gauze II agar (Appendix A), the activity was lost. For further studies, tests were performed after sowing with dots (Figure 1c, S14) or lines (Figure 1d and Appendix A). Some cultures were more suppressed than others (Appendix A). On average, the activity against fungi was more strongly expressed than antibacterial activity (ANOVA *p*-value = 0.003); however, other obtained fungi producing antimicrobial compounds were not suppressed (Figure 1(d6)). It is also possible that cultures obtained from the surfaces of bees are more resistant to the antimicrobial compounds produced (ANOVA *p*-value < 10^−4^). However, more testing is needed to draw definite conclusions about this.

The pooled microbiota sample was analyzed by the 16S and WGS sequencing methods (Figure 2). After sequencing, 2 063 042 WGS, 480795 V1-V2, and 192708 V3-V4 rRNA amplicon reads were obtained. The samples showed an atypically high proportion of unclassifiable sequences for 16S sequencing (48–55%), which may indicate the presence of unknown species or possible sample preparation problems. In contrast, 50% of the unclassified reads obtained from the metagenomics analysis were typical, so it supported the hypothesis of a higher portion of unknown species. Different databases were used for the annotations, all of which gave similar results when used in the analysis and did not help to solve the low classification rate problem (Appendix A).

The alpha diversity indices were similar to the indices calculated from the 16S sequencing data of V1-V2, V3-V4, and WGS (Shannon index: 1.59 ± 0.05, 1.41 ± 0.18, and 1.63; Simpson index: 0.70 ± 0.01, 0.68 ± 0.06, and 0.70, respectively). The alpha diversity was similar to that obtained in other articles (ANOVA *p*-value: 0.13) (Appendix A). For beta diversity, the annotation results differed from the rest of the pan-metagenome. This was due to the sample preparation, especially the enrichment of the sample.

Metagenomic annotation identified 23955 coding DNA sequences (CDSs), of which 10588 were able to identify product functions. In total, 37 genes (26 unique) for the biosynthesis of 22 different antimicrobial compounds (Appendix A) and 119 resistance genes (40 unique) were identified. Complete toxin–antitoxin systems, including higA/B and mqsA/R, were detected. 

Antimicrobial compounds possibly produced by the metagenome can vary in their origin, mechanism of action, and chemical nature; they may be small molecules or antimicrobial peptides, such as bacitracin and gramicidin. Each antimicrobial compound can possess either a narrow mechanism of action or a broad spectrum of antimicrobial activity.

Copper, cobalt–zinc–cadmium, and arsenic resistance cassettes were also detected. Several organic hydroperoxide resistance genes were found: ohrR (5), ohrB (3), and ohrA (2). The community was characterized by a wide spectrum of excreted and metabolizable secondary metabolites.

## 4. Discussion

The bee cuticle community grows on a relatively poor substrate. Studies with whole-bee 16S sequencing in their methodology are encountered but show little or no difference with gut sequencing [21,23]. Small values are shown by qPCR analysis and culture methods [15,16,52]. The bee surface microbiome is probably poor, with no more than a few thousand CFUs. The highest biomass is probably characteristic of mold fungi (in contrast with previous research), while the maximum biodiversity is probably among bacteria and yeasts migrating there from sugar-rich nectar and early honey (similar to previous studies [15,52]).

There are many studies on dead bee material, primarily in the context of the pathosphere [53,54,55]. This can be viewed as a successional series, the initial (yet stable) stages of which we are studying. In the course of this, the ecology of some microorganisms probably changes. In the transition to opportunism or parasitism of individual members of the community, such an environment becomes richer for a short time. Community succession is complete after bee death, and the final community is probably dominated by the mold fungi *Penicillium* and *Aspergillus*. The extent to which succession is sequential is unclear. It is also unclear what role the fungi, which are potential but not obligate pathogens [56,57], play in the cuticular microflora.

Metagenomic annotation revealed more than 20,000 coding sequences, half of them linked to specific functions. A significant portion was associated with the biosynthesis of various antimicrobial compounds, while numerous resistance genes were also identified. Some members of the community produced antimicrobial compounds. Whole-genome sequencing also confirmed the presence of genes for biosynthesis and resistance. The key producers of antimicrobial compounds may be Burkholderiaceae, Bacilli, and Actinobacteria. The detected components of toxin–antitoxin systems may also be involved as part of the community resistance maintenance system.

Fungicides and insecticides are known to affect this community [17]. The impact of these factors on its core species remains uncertain, given the varying levels of resistance observed in surface organisms.

The results of the metagenomic analysis differ from the known literature data on bee microbiology [2,15,22,41,58]. This may also be due to the fact that this study revealed features linked to ‘winter’ bees’ cuticular microbiome, a seasonal caste, integrating no-fertile individuals. It was characterized not only by changes in behavioral patterns but also in the microbial community profile, significantly different from that of foragers, nurses, and other castes. Such a community presented lower alpha diversity. The core groups were the main part, containing more pathogens by spring as a consequence of bee family depletion. Previously, 16S sequencing of bee cuticles was performed [15], which was reannotated in this work (Figure 3). The community was dominated by members of intestinal microflora, including obligately anaerobic *Lactobacillus*, *Bombilactobacillus*, and *Snodgrassella*, representing either technical contamination or the ubiquity of these organisms, which hardly play a real role in the cuticle community. The share in the microbiome of aerobes characteristic of the foregut and other hive environments—*Apilactobacillus* and Actinobacteria, probably represented by *Bifidobacterium asteroides*—was high. In our study, after enrichment, we found a different situation, but it is difficult to say which approach better describes the real, influential bee cuticular community.

The alpha diversity indices from the 16S and whole-genome sequencing were comparable and consistent with all studies. However, the beta diversity results varied across the pan-metagenome. Despite some similarities, there were also obvious differences with the intestinal microflora [2,20,26]. Unclassifiable species were more widely represented in the cuticle community and might play an important role in it. Key isolated actinomycete cultures remained unclassifiable by both culture and metagenomic methods. Their further study to confirm their role and persistence in bee cuticle communities is very important.

In contrast with bees, ants’ [13,14] and funnel-web spiders’ [4,12] cuticles have an ecological relationship with the soil microbiota. This was also evident in the observed species composition and total load. Current research indicates that there are probably no universal cuticular microorganisms of terrestrial arthropods. However, the same is true for the gut, although similar ecological groups are found there. Unlike the gut, we still do not really understand the structure and characteristics of surface microenvironments and the extent to which they are inhabited. More research is needed: both the physicochemical characterization of surface substrates and microbiological and metagenomic studies on various terrestrial insects. It is important to study not only adult stages but also larvae. Many experiments on bees are known, in which it is likely that if surface microflora is present, it is entirely inherited from the pupal or wax stage. At least it is not fatal, although full comparisons of bees with artificial and natural gut inoculation have not been made. All this emphasizes the importance of filling the research gap of cuticular microflora studies for basic and applied research.

## 5. Conclusions

Our work is one of the first studies on the bee cuticular community. Despite the application of various methods (16S sequencing, WGS, qPCR, and culture methods), the following questions still remain: what is the functionality of this community? What are the introduced species associated with this system, and are there core species? Our study supports the hypothesis that this community is independent, but unequivocal confirmation may come from analysis of the surface microflora by FISH or SEM. Metagenomic analyses and qPCR are hampered by significant contamination of the bee surface by representatives of their gut microbiome, which are obligate anaerobes. There is probably a higher proportion of actinomycetes and unclassifiable organisms in the community than in the bee gut. Their source may be the honeycomb of the hive, but the opposite situation may also be true, and, finally, different actinomycetes and unclassifiable organisms may be present on the surfaces of the hive and bees. This question requires further research.

Our results open a new chapter in micro-“*bee*”ology. Currently, there are no fully understood hive environments from a metagenomic perspective. The beehive, functioning as a superorganism, incorporates the genomes of its associated organisms into a unified hologenome, collectively responding to external stimuli. Given the strong links between the cuticle and gut microbiome of the bee and their indispensable role in nutrition and protection against pathogens, it would be interesting to study the dynamics of the cuticular microbial community throughout the year, especially during the demise season (when the communities undergo significant changes). Such data may allow the development of new techniques to prevent the spread of pathogens during the period when families are most vulnerable to pathogens.

## Figures and Tables

**Figure 1 biology-14-00088-f001:**
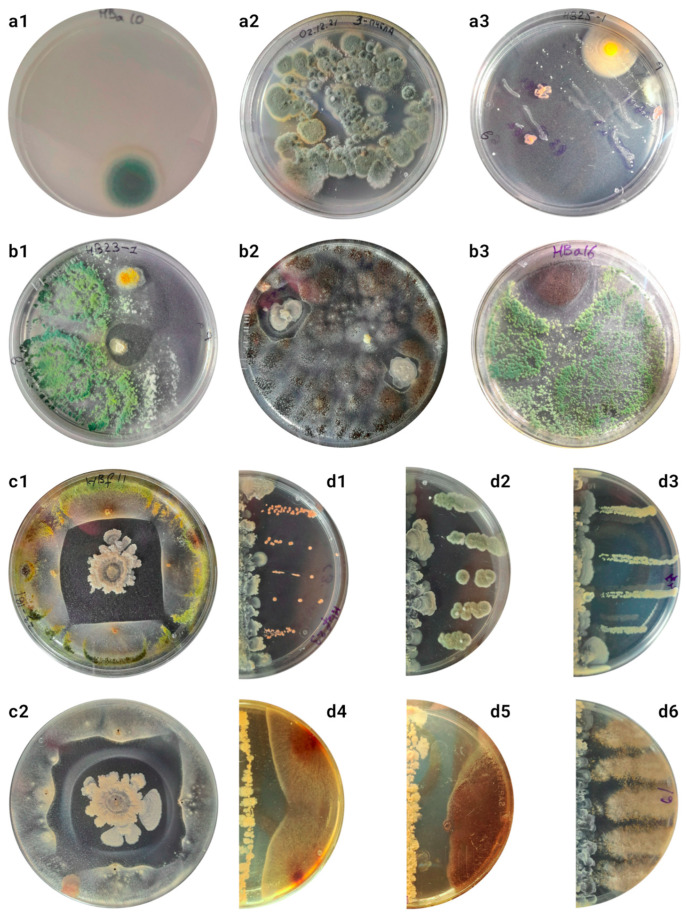
(**a1**–**a3**) Cultures of initial flushes from different sources. Photographs were taken 5 days after seeding. More detailed descriptions and cultures are shown in S1–S2. (**a1**) Cultures on solid Chapek’s medium without enrichment of cultures from a container in which the bee was kept for 1 h. Mean CFU: 8.8 ± 4.6. (**a2**) Cultures of total flushes from the surfaces of bees without enrichment on solid Czapek’s medium. Mean CFU: 130.2 ± 16.8. (**a3**) Cultures after enrichment on liquid medium for 24 h. Seeding by thinning stroke. (**b1**–**b3**) Examples of occurrence of suppression zones in initial cultures seed after enrichment. (**b1**,**b2**) Growth suppression by S10b culture. (**b3**) Growth suppression by S13a culture. (**c1**,**c2**,**d1**–**d6**) Tests for antibiotic production by S10b culture. More tests are presented in S14–15. (**c**) Spot test. (**c1**) Suppression of culture S13a. (**c2**) Suppression of culture S5c (photo at bottom). A “gallo” of unknown nature is visible, unique to the interaction of these two cultures. (**d1**–**d6**) Tests after seeding with lines. (**d1**) Interaction with S10c culture. Suppression zone: 0.38 ± 0.07 cm. (**d2**) Interaction with S5e culture. Suppression zone: 0.84 ± 0.14 cm. (**d3**) Interaction with S5b culture. Suppression zone: 0.28 ± 0.03 cm. (**d4**) Interaction with culture S5c. Suppression zone: 0.77 ± 0.03 cm. (**d5**) Interaction with culture S5f. Suppression zone: 0.84 ± 0.04 cm. (**d6**) Interaction with culture S10f. No suppression zone is detected.

**Figure 2 biology-14-00088-f002:**
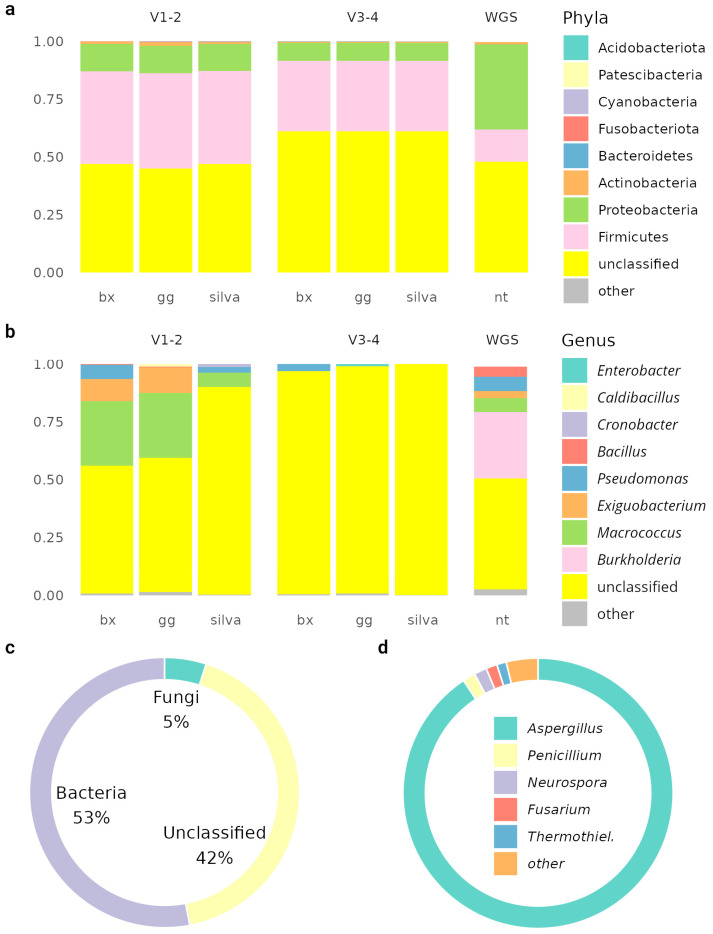
Results of metagenomic annotations. Classification results of bacterial reads at phylogroup (**a**) and genus (**b**) levels. For 16S sequencing data annotation, unweighted classifier training on SILVA, BeeXact (bx), and GreenGenes (gg) databases were used with combined samples with pan-metagenomic data. (**c**) Composition of sample using WGS metagenomics. (**d**) Composition of fungi at the genus level according to the results of WGS annotation.

**Figure 3 biology-14-00088-f003:**
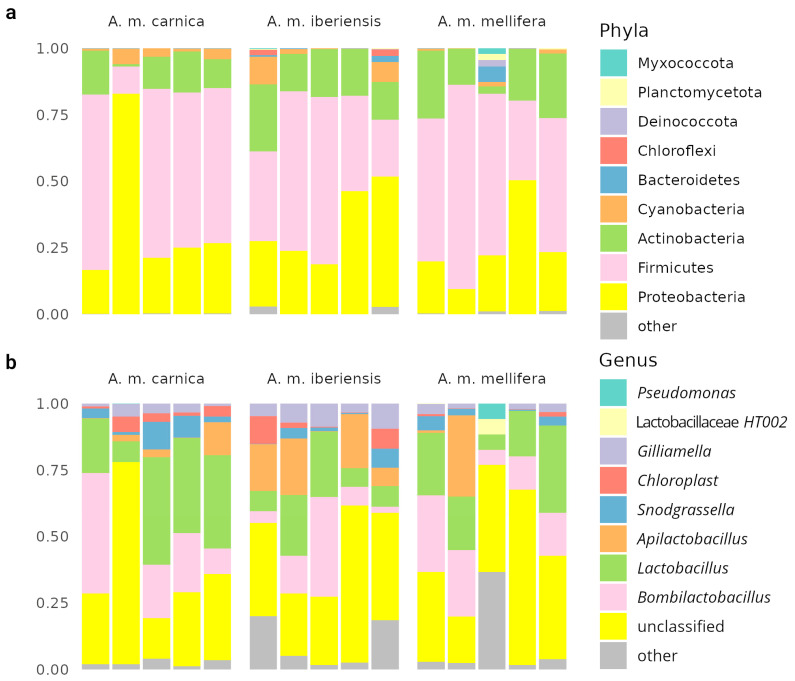
Compositions of the honeybee cuticular samples. Annotations on SILVA database at phylogroup (**a**) and genus (**b**) levels. Data from BioProject PRJNA879967.

## Data Availability

The sequencing data are available in BioProjects PRJNA1048732 and PRJNA1207523.

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
