# Peer review of "Investigating Aerobic Hive Microflora: Role of Surface Microbiome of Apis Mellifera"

_biology, 2025, doi:10.3390/biology14010088_

Round 1

Reviewer 1 Report

Comments and Suggestions for Authors

Dear Authors,

I find this paper very interesting and innovative. It would be significant to continue this contribution by analysing the products of the hive: honey and bee bread. On the other hand, it would also be relevant to study the effect of agrochemicals on the life of bees.

Few observations were made in the attached file of the manuscript. 

- Line 88-89: This sentence is not necessary. If you incorporate it, you should include further information about the experiments mentioned. 

- Regarding the methodology used, I am not a specialist in the topics of microbiology and genetics. However, it seems to me that it includes a thorough explanation but I leave it in the hands of the other evaluator if these are the appropriate techniques. 

- Bibliography cited twice: lines 365 and 372. 

English is not my native language, for that reason I do not comment on the writing of the manuscript.

Kind regards,

Author Response

Dear reviewer #1, 

We greatly appreciate the time and effort you have dedicated to reviewing our manuscript, and we would like to thank you for your constructive comments which have brought an important issue to our attention.

We are extremely grateful for your positive feedback regarding our research study. We are delighted that you find the paper interesting and innovative.We would like to express our sincere appreciation for your suggestion to extend this study by analysing hive products such as honey and bee bread, as well as the potential impact of agrochemicals on the life of bees.Both of these directions are highly relevant and would significantly contribute to a more comprehensive understanding of the topic.We will carefully consider incorporating these aspects in our future investigations.

# Com. 1. Line 88-89: This sentence is not necessary. If you incorporate it, you should include further information about the experiments mentioned.

Thanks for the comment. We were referring to the experiments described next. A note has been added to the text.

# Com. 2. Regarding the methodology used, I am not a specialist in the topics of microbiology and genetics. However, it seems to me that it includes a thorough explanation but I leave it in the hands of the other evaluator if these are the appropriate techniques. 

We would like to express our gratitude for your feedback regarding the methods section of our paper. We acknowledge that microbiology and genetics may not fall within your primary area of expertise.

We appreciate your acknowledgement that the explanation provided appears thorough. We would like to assure you that we welcome the insights of the other evaluator to determine the appropriateness of the techniques employed.

# Com. 3. Bibliography cited twice: lines 365 and 372.

We would like to express our gratitude for highlighting this matter. A thorough review has been conducted, resulting in the identification and resolution of the issue concerning the bibliography citation, which was present on two occasions, at lines 365 and 372. The document has been updated to reflect these changes.

We are confident that these revisions have significantly improved the manuscript and hope that the changes meet with your approval. Please do not hesitate to contact us should there be any additional aspects of the manuscript that require clarification. Thank you again for your insightful comments and assistance in refining our paper. 

Sincerely yours,

Daniil Smutin, Adonin Leonid and co-authors.

Reviewer 2 Report

Comments and Suggestions for Authors

The authors investigated the cuticular microbial community of honey bees and combined multiple methods, such as sequencing (16s and wgs) and cultivation, to do this. Moreover, the cultivates were tested for their antimicrobial properties. With that, they provide a great and novel overview of the cuticular microbial community. This is a great addition to the already existing data on the gut community and will hopefully inspire more research on the role of the cuticular community. 

So, overall, I enjoyed the manuscript. However, all parts need some reworking. I completely missed the point that the authors worked with winter bees. However, this is a significant point, as this can influence the findings a lot. The authors need to address this during their introduction and include it in the discussion. Also, some parts of the experiment are missing; for example, they looked at different body parts and explained why this was done.

Moreover, the material and method parts, as well as the discussion and conclusion, need some rework. 

The material and method part would greatly benefit from a few sentences preparing the reader for what is to come and a more evident separation/structuring of the different aspects of the study. 

The discussion lacks a bit of depth in the current status. The authors sort of throw in some assumptions/statements but don't elaborate on what basis they made these assumptions and what they mean. Some more in-depth elaborations are needed, so how did they come to their assumption, and what conclusion do they draw from it? The conclusion would benefit from an outlook on what would be interesting to do next.

Abstract: In general, mention that you work with winter bees, as this could hugely influence your data

Lines 24-25: I don't see how you can make this statement, especially the second part. Remove everything after the "and". 

Introduction:

Line 39: What did they find? Why are they a good model? If they are a good model, why didn't you use them? Rephrase.

Line 42-43:Also nestmate recognition in ants via CHCs produced from microbes on the cuticula.

Line 43-44: You need to elaborate on this further, as I would argue the main barrier is the physical barrier of the cuticula, and the microbes are a plus. Moreover, many insects use various secretions to manage their cuticula. Elaborate in text.

Line 46: What do you mean by "honeybees's environmental microbes"? Do you mean their nest microbiome or their surroundings? Please clarify.

Line 48-49: What do you want to say in this sentence? Try to clarify within the text.

Line 61-69: Different font and or/ text size.

Line 67-68: Elaborate how this could benefit the bee within the text.

Line 76-78: I would argue that you provide a great baseline for future in-depth studies, as right now, you only screen for presence and, in some, for antimicrobial properties. However, from your research, there is no possibility of concluding whether the microbes are important or not. Maybe think about changing it.

Methods: You have a great design, but the descriptions are sometimes challenging to follow. For example, what cultures were used in the antibiotic test? Are they the ones from body parts, the pure cultures, or both? Try to clarify. Also, a short, general explanation of the setup would be beneficial.

Line 84: I'm not entirely sure about your sample size. Did you use five bees per hive, so the sample size is 15? Or was it only five bees in total? The sample size is not apparent. Fix within the text.

Line 86: Which body parts? When were they separated? Why do you look at different body parts? You should introduce this in the intro.

Line 94: what crops?

Line 103: You mention a "first", but where is the second?

Line 113-114: Again, I was missing context; see comment made during the first time reading: "So still not clear which parts did you use, also why do you think they differ and why is it the case that they differ"

Line 121-122: Please clarify what cultures were used. I was a bit lost.

Results: Maybe try to separate the results and figures a bit more according to the different "experiments".

Line 168: So these are the results from the shedding experiment?

Line 196: It would be good if the authors mention table S1 in the methods more often to point out that they compare different media.

Line 203: Why winter bees, will be very different in summer, you have to mention this in the methods and abstract, maybe already in the intro and discuss this in the "discussion" part. 

Line 224: How many reads in total? And what percentage were unknown? Please add to the text.

Line 241: Please spell "CDSs" out when using it for the first time.

Discussion: As mentioned above, try to elaborate more on the points given. Also, include a discussion on the fact that you used winter bees. 

Line 254-255: What do you mean here? Elaborate within the text.

Line 262-266: why do you bring this here? I'm missing a bit of the context. Try to clarify what point you want to make here.

Line 277: What do you mean by "bark representatives"?

Line 276-278: Again, elaborate on why you give this argument.

Line 281-282: On what basis do you determine them as contamination? Isn't it quite likely they are in the environment of bees and with that on their cuticula? Especially as the microbiome is established after hatching through oral feeding, and anaerobe microbes can survive in biofilms.

Line 294: On what basis do you think the unclassified are important? Please elaborate on why you think this and for what they could be important in the text.

Conclusion:

Line 302: What do you mean by "bark species"?

Line 304: Also, analysing the community based on RNA extraction would be interesting, as by using DNA, we also screen for dead and/or inactive microbes, and it would be beneficial to have a measure where we focus only on active microbes, aka via RNA. 

Line 305: Again you don't know if they are contaminations, also just because they are anaerobe doesn't mean that they can't survive as biofilms can provide an anaerobe environment for them.

Line 309: Why do you think they are unclassifiable, at least for bacteria? Something like genus or family should be possible, or do you mean up to species level?

Line 314: It would be nice to end with an outlook. So what would you suggest to do?  What would be interesting to do next?

Author Response

Dear reviewer #2, 

We would like to express our sincere gratitude for your gracious and supportive remarks concerning our research endeavours. We deeply value your discerning feedback and are delighted that you perceive our study as a meaningful contribution to the field.

We are genuinely gratified to learn that our research on the cuticular microbial community has resonated with you and that you regard it as a pioneering addition to the existing literature. It is our fervent hope that this research will serve as a catalyst for further exploration into the functions of these microbial communities.

# Com. 1. So, overall, I enjoyed the manuscript. However, all parts need some reworking. I completely missed the point that the authors worked with winter bees. However, this is a significant point, as this can influence the findings a lot. The authors need to address this during their introduction and include it in the discussion. Also, some parts of the experiment are missing; for example, they looked at different body parts and explained why this was done. 

We are extremely grateful for your meticulous feedback and for highlighting the significance of working with winter bees. We concur that this is a substantial aspect, and we will ensure that it is explicitly mentioned in the "Methods" section of the manuscript.Furthermore, we will address the influence of working with winter bees in both the introduction and discussion to provide more contextual background for our findings. 

# Com. 2. Moreover, the material and method parts, as well as the discussion and conclusion, need some rework. The material and method part would greatly benefit from a few sentences preparing the reader for what is to come and a more evident separation/structuring of the different aspects of the study. 

Thank you for your comment. We discussed with co-authors even before publication the splitting of methods and concluded that a large split would be rather confusing and probably more necessary for reproducing the results of the paper than for its global understanding. Therefore, for convenience, we have divided the methods into large chapters and added an explanatory commentary following your advice. We hope this will improve the reader's experience.

# Com. 3. The discussion lacks a bit of depth in the current status. The authors sort of throw in some assumptions/statements but don't elaborate on what basis they made these assumptions and what they mean. Some more in-depth elaborations are needed, so how did they come to their assumption, and what conclusion do they draw from it? The conclusion would benefit from an outlook on what would be interesting to do next.

Thank you for your comment. Unfortunately, it is not only our conclusion that suffers from a lack of disclosure, but the scientific field as a whole. Despite the comprehensive study of the intestinal metagenome of bees, the rest of the hive microbiome (and even the microbiome of their surface) has hardly been studied before, even by purely cultural methods. Our paper seeks to close the research gap: we do not understand the true reason for the lack of materials on the cuticular metagenomes of mockingbirds. Undoubtedly, they are less abundant compared to mammals, they have less bacterial load, and they are more difficult to study. In addition there are other hypotheses discussed in the introduction. The aim was to use relevant comparisons of results obtained from other social insects or from other hive environments. The study was methodically designed to extend the research in this area.

# Com. 4. Abstract: In general, mention that you work with winter bees, as this could hugely influence your data

We have taken your comment into account. Information about winter bee handling is now in the sections “Abstract” and “Methods”.

# Com. 5. Lines 24-25: I don't see how you can make this statement, especially the second part. Remove everything after the "and". 

We thank you for pointing out the overstatement in lines 24-25. We have therefore removed the problematic portion of the sentence and revised the initial part for clarity.

# Com. 6. Line 39: What did they find? Why are they a good model? If they are a good model, why didn't you use them? Rephrase.

Funnel-web spiders provide a valuable model for studying host-microbe interactions on the exoskeleton due to their complex ecological interactions with environmental microbes, which influence both host physiology and behavior. Studies have shown that bacterial exposure of copulatory organs can affect mating behavior [1] and that exposure to specific cuticular bacteria can alter foraging aggressiveness [2]. Such findings highlight the significant role surface microbiome play in shaping these animal physiology and behavior and underscore the importance of understanding these interactions, including wider host-microbe dynamics. 

  1. Spicer, M.E.; Pruitt, J.N.; Keiser, C.N. Spiders, Microbes and Sex: Bacterial Exposure on Copulatory Organs Alters Mating Behaviour in Funnel‐web Spiders. Ethology 2019, 125, 677–685, doi:10.1111/eth.12921.
  2. Parks, O.B.; Kothamasu, K.S.; Ziemba, M.J.; Benner, M.; Cristinziano, M.; Kantz, S.; Leger, D.; Li, J.; Patel, D.; Rabuse, W.; et al. Exposure to Cuticular Bacteria Can Alter Host Behavior in a Funnel-Weaving Spider. Curr. Zool. 2017, doi:10.1093/cz/zox064.

# Com. 7. Line 42-43:Also nestmate recognition in ants via CHCs produced from microbes on the cuticula.

It is regrettable that such a significant topic has been overlooked in our work. To address this omission, the text has been augmented with data on the contribution of cuticular microbiota to CHC synthesis, along with an examination of the role these compounds play in Hymenoptera.

# Com. 8. Line 43-44: You need to elaborate on this further, as I would argue the main barrier is the physical barrier of the cuticula, and the microbes are a plus. Moreover, many insects use various secretions to manage their cuticula. Elaborate in text.

We completely agree with you, especially since the actual role of cuticular microflora in insects is not fully established - only hypotheses. It is corrected in the text.

# Com. 9. Line 46: What do you mean by "honeybees's environmental microbes"? Do you mean their nest microbiome or their surroundings? Please clarify.

Speaking of bee microbiology, the term ‘environmental microorganisms’ is related to any hive environments, such as honey, beebread, wax, combs, etc. For clarification we add ‘hive’ to text

# Com. 10. Line 48-49: What do you want to say in this sentence? Try to clarify within the text.

Thank you for this comment; the sentence is rephrased.

# Com. 11. Line 61-69: Different font and or/ text size.

We are grateful to you for highlighting the issue concerning the font and text size in lines 61-69. This has now been rectified.

# Com. 12. Line 67-68: Elaborate how this could benefit the bee within the text.

Exactly  Aureobasidium pullulans may not influence bees, but others could possibly influence their immunity. Corrected.

# Com. 13. Line 76-78: I would argue that you provide a great baseline for future in-depth studies, as right now, you only screen for presence and, in some, for antimicrobial properties. However, from your research, there is no possibility of concluding whether the microbes are important or not. Maybe think about changing it.

Thank you for this comment, we agree with it. At the same time, in data-driven metagenomic studies, often confirming the persistence or at least the number of certain microorganisms is often the starting point for identifying their role. Researchers proceed from the logic: if something is there, it is needed for something. We not only show the presence, but also indicate possible mechanisms for the realization of immune and barrier functions, without considering only the role in the formation of the surface composition, which may be the goal of further research.

# Com. 14. Methods: You have a great design, but the descriptions are sometimes challenging to follow. For example, what cultures were used in the antibiotic test? Are they the ones from body parts, the pure cultures, or both? Try to clarify. Also, a short, general explanation of the setup would be beneficial.

Thanks for the comment. We have added an explanation before the start of the methods that should clarify this.

# Com. 15. Line 84: I'm not entirely sure about your sample size. Did you use five bees per hive, so the sample size is 15? Or was it only five bees in total? The sample size is not apparent. Fix within the text.

Thanks for the comment; fixed.

# Com. 16. Line 86: Which body parts? When were they separated? Why do you look at different body parts? You should introduce this in the intro.

Thanks for the comment; added.

# Com. 17. Line 94: what crops?

Thanks for the comment. We meant pure cultures; fixed.

# Com. 18. Line 103: You mention a "first", but where is the second?

Thank you for your comment. The logic of the paragraph was originally structured differently, and the “first” was retained from there. Fixed.

# Com. 19. Line 113-114: Again, I was missing context; see comment made during the first time reading: "So still not clear which parts did you use, also why do you think they differ and why is it the case that they differ"

Thank you for your comment. Before researching, we weren't sure if they were different. Serious differences in microflora on different parts of the bee may show both contamination and the presence of different microenvironments on their surface. Added to the text.

# Com. 20. Line 121-122: Please clarify what cultures were used. I was a bit lost.

Thank you for the comment. We added this information at the start of the Methods section.

# Com. 21. Results: Maybe try to separate the results and figures a bit more according to the different "experiments".

We are grateful for the proposal regarding the arrangement of the results section and welcome the constructive criticism. A thorough examination of the presentation of the results and figures of the experiments has been conducted, and it is believed that the present structure offers a cohesive and integrated narrative that successfully communicates the study's overarching conclusions.

We have striven to present the results in a clear and logical manner, adhering to a style that we feel is well-suited for the intended audience and consistent with the overall stylistic conventions of the journal. However, we will certainly take your suggestion into consideration for future revisions and in the presentation of our work going forward.We value your input and remain open to further suggestions that may improve the clarity of our manuscript.

# Com. 22. Line 168: So these are the results from the shedding experiment?

Yes, we added this information for the clarification.

# Com. 23. Line 196: It would be good if the authors mention table S1 in the methods more often to point out that they compare different media.

Thank you for the comment; added here and in the antibiotic production studies section. 

# Com. 24. Line 203: Why winter bees, will be very different in summer, you have to mention this in the methods and abstract, maybe already in the intro and discuss this in the "discussion" part. 

It is acknowledged that there exist substantial disparities between the physiological and behavioural characteristics of winter bees and those of summer bees, a pivotal factor in the interpretation of our findings.We would like to acknowledge and welcome the recommendation to explicitly state the use of winter bees throughout the manuscript. We have already noted the use of winter bees specifically in the "Methods" section (line 203), however, we have now also made it more prominent in the abstract and introduced the rationale for this choice in the introduction.

# Com. 25. Line 224: How many reads in total? And what percentage were unknown? Please add to the text.

Thanks for the comment; added.

# Com. 26. Line 241: Please spell "CDSs" out when using it for the first time.

Thanks for the comment; added.

# Com. 27. Discussion: As mentioned above, try to elaborate more on the points given. Also, include a discussion on the fact that you used winter bees. 

Thanks for the comment; added.

# Com. 28. Line 254-255: What do you mean here? Elaborate within the text.

We would like to take this opportunity to clarify our statement. The 16S rRNA gene sequencing methodology often involves DNA extraction following whole-bee homogenization rather than from specific anatomical regions like the gut (commonly examined area) or other body tissues. This approach results in a dataset primarily consisting of major gut-associated taxa, and it may overlook variations in microbial communities across others organism parts. To further illustrate this point, we have added citations to several pertinent publications that employed whole-bee 16S rRNA sequencing in the revised version of the manuscript.

# Com. 29. Line 262-266: why do you bring this here? I'm missing a bit of the context. Try to clarify what point you want to make here.

Thanks for your comment. The point is that there are many studies of bee cadaveric material, primarily in the context of the pathosphere. This can be viewed as a successional series, the final community of which is well studied. Added context to the text.

# Com. 30. Line 277: What do you mean by "bark representatives"?

The term “bark representatives” was entered erroneously, the revised manuscript now employs the term “core species”.

# Com. 31. Line 276-278: Again, elaborate on why you give this argument.

Thanks for the comment; added to the text. In general, fungicides and pesticides severely affect bee health. This was previously attributed primarily to their effect on the gut microflora; however, it is clear that their effects are not limited to this community. We thus emphasize an important research gap.

# Com. 32. Line 281-282: On what basis do you determine them as contamination? Isn't it quite likely they are in the environment of bees and with that on their cuticula? Especially as the microbiome is established after hatching through oral feeding, and anaerobe microbes can survive in biofilms.

We appreciate you highlighting the important issue of classifying certain microorganisms as contaminants. We probably have some variation of the term “contamination”. Here we did not mean technical contamination, but contamination from the external environment; when certain organisms are introduced into an environment in which they probably do not live, but are detected by various methods. For clarification, we added this information to the text

We concur that microbial communities can form biofilms on various surfaces, however, we believe this is unlikely to be a significant factor on the constantly renewing cuticular surface of bees, particularly given the identity of the microorganisms. Please see our response to comment #34 for a detailed discussion of biofilms.

# Com. 33. Line 294: On what basis do you think the unclassified are important? Please elaborate on why you think this and for what they could be important in the text.

Thanks for the comment, it is a great point. We have softened the wording; we have this idea because we encounter unclassifiable actinomycetes, which are often producers of antimicrobial compounds.

# Com. 34. Conclusion: Line 302: What do you mean by "bark species"?

We are indebted to for highlighting the error. This was due to a regrettable typographical mistake, and the term has now been corrected to "core species" in the revised manuscript.

# Com. 35. Line 304: Also, analysing the community based on RNA extraction would be interesting, as by using DNA, we also screen for dead and/or inactive microbes, and it would be beneficial to have a measure where we focus only on active microbes, aka via RNA. 

We are grateful for the constructive comments that have been provided regarding the analysis of the community based on RNA extraction. It is acknowledged that a focus on RNA would provide a more accurate measurement of the active microbial community, as it circumvents the limitations of DNA analysis by eschewing the inclusion of dead and/or inactive microbes.

This approach will be given due consideration in future research endeavours, with the objective of achieving a more precise understanding of the active microbial populations.

# Com. 36. Line 305: Again you don't know if they are contaminations, also just because they are anaerobe doesn't mean that they can't survive as biofilms can provide an anaerobe environment for them.

We acknowledge that the presence of anaerobic bacteria does not definitively preclude their survival within biofilms, even in environments that are not strictly anaerobic. However, we would like to emphasise that the cuticle is a highly dynamic structure, undergoing continuous renewal and shedding. This dynamic nature suggests that the surface of the cuticle is unlikely to provide a stable and conducive environment for the formation and sustained growth of microbial biofilms.While we cannot definitively rule out the presence of transient or localized biofilms, given the continual renewal process, we suspect that the observed anaerobic bacteria are more likely contaminants rather than established biofilm communities on the cuticle's surface.

The discussion has been amended to reflect this nuance and acknowledge the possibility of localized biofilm formation, while maintaining the distinction between these observations and potential persistent colonization within a dynamic structure like the cuticle.Your feedback is invaluable in improving the precision and accuracy of our interpretation.

# Com. 37. Line 309: Why do you think they are unclassifiable, at least for bacteria? Something like genus or family should be possible, or do you mean up to species level?

In the community, about 50% of the sequences fail to be classified at any level of classification, which seems rather unusual (especially for 16S), here they were meant.

# Com. 38. Line 314: It would be nice to end with an outlook. So what would you suggest to do? What would be interesting to do next?

We are grateful for the suggestion to incorporate an outlook in the manuscript, and we intend to emphasise the significance of seasonal dynamics in the cuticular microbiome in future studies. This area is of particular interest and will be explored in subsequent research. The aim is to investigate how the microbiome composition may vary across different seasons and its potential impact on bee health. This could provide valuable insights into the broader ecological context of the findings and facilitate a more comprehensive understanding of the role of the microbiome in honeybee biology. This direction will be emphasised in the manuscript's conclusion.

We are confident that these revisions have significantly improved the manuscript and hope that the changes meet with your approval. Please do not hesitate to contact us should there be any additional aspects of the manuscript that require clarification. Thank you again for your insightful comments and assistance in refining our paper. 

Sincerely yours,

Daniil Smutin, Adonin Leonid and co-authors.

Reviewer 3 Report

Comments and Suggestions for Authors

Dear Editor,

This research investigate the composition of microbial communities on the surface of honeybees and their functional roles within honeybee biology. The study adopts an approach that involves extracting microbial contents from various components of bee hives and different organ tissues on the honeybee’s surface, culturing these extracts in a nutrient medium, and subsequently analyzing the resulting samples through 16S rRNA sequencing. This methodology yields a comprehensive set of data.

Given the extensive array of tissues on the honeybee’s surface that potentially harbor microbial communities, the study’s efforts predominantly focus on bacterial cultivation. The findings reveal a detailed composition of microbial colonies on the honeybee’s surface. However, the functional roles and origins of these bacterial and fungal communities remain unresolved. While the manuscript presents speculative discussions on these aspects, these hypotheses necessitate further experimental substantiation for refinement and validation.

Below, I provide my detailed review comments:

 1. In the keywords section of this manuscript, can the keyword Honey bee Apis mellifera be modified to either Honey bee or Apis mellifera? The author can refer to articles about bees published by the journal for guidance.

 2. There are too many keywords, please reduce them appropriately.

 3. In lines 37-43 of the introduction section of this manuscript, the author mentions a model of the interaction between funnel-web spiders and the microorganisms on their exoskeletons, and also discusses the strong correlation between ant size and the number of microorganisms on their body surfaces. Could the author provide specific examples to elaborate on the interactions between invertebrates and the microorganisms on their body surfaces, to facilitate a deeper understanding for readers? Overall, the clarification of the research background is still lacking.

 4. What is the significance of the sentence in lines 44-45 of the introduction section of this manuscript? It seems unrelated to the content of this paragraph. This sentence could be moved to the beginning of the paragraph to introduce the existence of microbial communities on insect body surfaces.

 5. Is the elaboration on the interaction between bees and the microbial communities on their body surfaces comprehensive in lines 46-51 of the introduction section? The content seems somewhat vague.

 6. Is the fourth paragraph of the introduction section of this manuscript the result of this study, or is it the result of previous research? If it is previous research, could the third and fourth paragraphs be combined into one, serving as a clarification of the research background on the interaction between bees and the microorganisms on their body surfaces?

 7. Are there any typographical errors in lines 63-69 of the introduction section of this manuscript? If so, please correct them.

 8. Could the limitations of this study be explained?

 9. In Figure 1, the caption format for (a1) and (a2) is not consistent with that for (a3), as a period (.) is added after (a3).

 10. Figure 1 (a3) shows the result of a strain enriched in liquid medium for 24 hours before plating. According to the usual pattern, high CFU values and visually abundant photos should be obtained. Why are the experimental results as shown in Figure 1 (a3)?

 11. Do V1-2 and V3-4 in Figure 2 represent pooled samples? Could a detailed explanation be provided?

 12. Figure 3b shows the 16S sequencing results at the genus level after cultivating the microbial community of bee cuticles. The consistency among replicates of each bee species is poor. What are the possible reasons for this?

 13. In the discussion section, the author states that the microbial community of bee cuticles is similar to that of the gut. What is the connection or reason for this similarity?

 14. Does Figure 3 show sequencing results obtained from five bees collected from each of three hives? Is it necessary to collect bees from multiple hives for 16S sequencing to be representative for each bee subspecies? That is, to provide at least three hive replicates for each bee subspecies.

 15. In the discussion section, the author points out that the microbial community on the surface of bees plays a role in resisting external stresses, such as producing antibacterial compounds.

 16. In the supplementary materials, how many independent samples of bee tissue washings were placed in each culture dish?

 17. In Figure S17 of the supplementary materials, why is the number of wave points in the first row of the SILVA optimal feature results different from the other two databases?

Author Response

Dear reviewer #3, 

This research investigate the “composition of microbial communities on the surface of honeybees and their functional roles within honeybee biology.” The study adopts an approach that involves extracting microbial contents from various components of bee hives and different organ tissues on the honeybee’s surface, culturing these extracts in a nutrient medium, and subsequently analyzing the resulting samples through 16S rRNA sequencing. This methodology yields a comprehensive set of data.

Given the extensive array of tissues on the honeybee’s surface that potentially harbor microbial communities, the study’s efforts predominantly focus on bacterial cultivation. The findings reveal a detailed composition of microbial colonies on the honeybee’s surface. However, the functional roles and origins of these bacterial and fungal communities remain unresolved. While the manuscript presents speculative discussions on these aspects, these hypotheses necessitate further experimental substantiation for refinement and validation.

We would like to express our sincere gratitude for the comprehensive and meticulous evaluation of our manuscript. We deeply value the time and discerning feedback provided, which we firmly believe will substantially enhance the calibre of our work.We acknowledge your observations concerning the breadth of our study and the necessity for additional substantiation of the functional roles of the microbial communities we have identified. While we have furnished a thorough account of the composition of these communities, it is evident that further experimental endeavours are imperative to fully elucidate their functional roles and origins.

We have given full consideration to all points raised and are committed to addressing each one thoroughly. In response, we have made specific revisions to the manuscript, where we have clarified certain aspects of our methods and added further context to the discussion. Our detailed responses to your key comments are provided below and are highlighted in yellow within the text of the revised manuscript.

# Com. 1. In the keywords section of this manuscript, can the keyword “Honey bee Apis mellifera” be modified to either “Honey bee” or “Apis mellifera”? The author can refer to articles about bees published by the journal for guidance.

Your observation regarding the keywords is appreciated. The keyword "Honey bee Apis mellifera" has been revised, and it is now retained solely in its Latin binomial form, i.e. "Apis mellifera". We are grateful to you for highlighting the journal's preferred style for keywords.

# Com. 2. There are too many keywords, please reduce them appropriately.

A more appropriate selection of keywords has been made, with the number being reduced to those terms that best represent the scope and focus of the study. The revised list of keywords is as follows: Apis mellifera, 16S, whole metagenomic sequencing, bacterial diversity, cuticular communities, and symbiosis. It is believed that this revised set of keywords will provide a concise and accurate representation of the work for indexing and search purposes.

# Com. 3. In lines 37-43 of the introduction section of this manuscript, the author mentions a model of the interaction between funnel-web spiders and the microorganisms on their exoskeletons, and also discusses the strong correlation between ant size and the number of microorganisms on their body surfaces. Could the author provide specific examples to elaborate on the interactions between invertebrates and the microorganisms on their body surfaces, to facilitate a deeper understanding for readers? Overall, the clarification of the research background is still lacking.

Funnel-web spiders provide a valuable model for studying host-microbe interactions on the exoskeleton due to their complex ecological interactions with environmental microbes, which influence both host physiology and behavior. Studies have shown that bacterial exposure of copulatory organs can affect mating behavior [1] and that exposure to specific cuticular bacteria can alter foraging aggressiveness [2]. Such findings highlight the significant role surface microbiome play in shaping these animal physiology and behavior and underscore the importance of understanding these interactions, including wider host-microbe dynamics. 

  1. Spicer, M.E.; Pruitt, J.N.; Keiser, C.N. Spiders, Microbes and Sex: Bacterial Exposure on Copulatory Organs Alters Mating Behaviour in Funnel‐web Spiders. Ethology 2019, 125, 677–685, doi:10.1111/eth.12921.
  2. Parks, O.B.; Kothamasu, K.S.; Ziemba, M.J.; Benner, M.; Cristinziano, M.; Kantz, S.; Leger, D.; Li, J.; Patel, D.; Rabuse, W.; et al. Exposure to Cuticular Bacteria Can Alter Host Behavior in a Funnel-Weaving Spider. Curr. Zool. 2017, doi:10.1093/cz/zox064.

# Com. 4. What is the significance of the sentence in lines 44-45 of the introduction section of this manuscript? It seems unrelated to the content of this paragraph. This sentence could be moved to the beginning of the paragraph to introduce the existence of microbial communities on insect body surfaces.

Thanks for the comment; reworded the paragraph.

# Com. 5. Is the elaboration on the interaction between bees and the microbial communities on their body surfaces comprehensive in lines 46-51 of the introduction section? The content seems somewhat vague.

Thanks for the comment; updated.

# Com. 6. Is the fourth paragraph of the introduction section of this manuscript the result of this study, or is it the result of previous research? If it is previous research, could the third and fourth paragraphs be combined into one, serving as a clarification of the research background on the interaction between bees and the microorganisms on their body surfaces?

Thank you for your insightful feedback regarding the structure of our introduction, specifically concerning paragraphs three and four (lines 46-69). We appreciate your suggestion to potentially combine these paragraphs. We have carefully considered your point and have revisited these sections with the aim of optimizing the flow of information.

We have opted to maintain their separation. We feel that merging these paragraphs would result in a more dense and potentially less accessible text for the reader, potentially hindering the clarity we have aimed for in our introduction.

We have carefully crafted these sections to provide the reader with a logical and well-paced entry into the specific area of investigation we present in our study. However, we welcome any further suggestions that could improve the overall clarity and impact of our introduction.

# Com. 7. Are there any typographical errors in lines 63-69 of the introduction section of this manuscript? If so, please correct them.

We are grateful to you for highlighting the issue concerning the font and text size in lines 63-69, and any possible typos. This has now been rectified.

# Com. 8. Could the limitations of this study be explained?

Yes, they are presented in detail in the discussion.

# Com. 9. In Figure 1, the caption format for (a1) and (a2) is not consistent with that for (a3), as a period (“.”) is added after (a3).

Thanks for the comment; fixed.

# Com. 10. Figure 1 (a3) shows the result of a strain enriched in liquid medium for 24 hours before plating. According to the usual pattern, high CFU values and visually abundant photos should be obtained. Why are the experimental results as shown in Figure 1 (a3)?

Thank you for your comment. Unfortunately, for the rest of the bees there are only photos at later stages, when secondary cultures have already appeared, so this one is presented.

# Com. 11. Do V1-2 and V3-4 in Figure 2 represent pooled samples? Could a detailed explanation be provided?

Yes, these are pooled samples; a detailed description of obtaining sequencing results is provided in the methods.

# Com. 12. Figure 3b shows the 16S sequencing results at the genus level after cultivating the microbial community of bee cuticles. The consistency among replicates of each bee species is poor. What are the possible reasons for this?

Differences in methodology and sequenced sites become key. Firmictues and Proteobacteria are identified significantly worse by V3-V4 sequencing, whereas we expected a somewhat different picture.

# Com. 13. In the discussion section, the author states that the microbial community of bee cuticles is similar to that of the gut. What is the connection or reason for this similarity?

Linkage is observed in species composition, and partly even in quantitative composition in samples obtained by direct sequencing. This may be due to contamination; updated for clarity in the text.

# Com. 14. Does Figure 3 show sequencing results obtained from five bees collected from each of three hives? Is it necessary to collect bees from multiple hives for 16S sequencing to be representative for each bee subspecies? That is, to provide at least three hive replicates for each bee subspecies.

Thanks for the comment. Unfortunately, the results are pooled samples, so different variants of the analysis are presented rather than replicates. We aimed to show the relevance of the methodology and evaluate its efficiency.

# Com. 15. In the discussion section, the author points out that the microbial community on the surface of bees plays a role in resisting external stresses, such as producing antibacterial compounds.

Thanks for the comment. Yes, and we showed it in the series of experiments. However, real antimicrobial production on the surface still should be proved.

# Com. 16. In the supplementary materials, how many independent samples of bee tissue washings were placed in each culture dish?

Thanks for the comment. It is mentioned only in methods; there was a single bee per cultural experiment.

We are confident that these revisions have significantly improved the manuscript and hope that the changes meet with your approval. Please do not hesitate to contact us should there be any additional aspects of the manuscript that require clarification. Thank you again for your insightful comments and assistance in refining our paper. 

Sincerely yours,

Daniil Smutin, Adonin Leonid and co-authors.

Round 2

Reviewer 2 Report

Comments and Suggestions for Authors

Thanks for the nicely done revision. 

Reviewer 3 Report

Comments and Suggestions for Authors The manuscript has been sufficiently improved; I don't have any more questions.